

# Identification of NUF2 and FAM83D as potential biomarkers in triple-negative breast cancer

Xiuming Zhai[1], Zhaowei Yang[2], Xiji Liu[1], Zihe Dong[3] and Dandan Zhou[3]

[1] Department of Laboratory Medicine, The Third Affiliated Hospital of Chongqing Medical University, Chongqing, China
[2] Department of Breast and Thyroid, Chongqing Hospital of Traditional Chinese Medicine, Chongqing, China
[3] Department of Laboratory Medicine, Chongqing Hospital of Traditional Chinese Medicine, Chongqing, China

## ABSTRACT

**Background**. Breast cancer is a heterogeneous disease. Compared with other subtypes of breast cancer, triple-negative breast cancer (TNBC) is easy to metastasize and has a short survival time, less choice of treatment options. Here, we aimed to identify the potential biomarkers to TNBC diagnosis and prognosis.

**Material/Methods**. Three independent data sets (GSE45827, GSE38959, GSE65194) were downloaded from the Gene Expression Omnibus (GEO). The R software packages were used to integrate the gene profiles and identify differentially expressed genes (DEGs). A variety of bioinformatics tools were used to explore the hub genes, including the DAVID database, STRING database and Cytoscape software. Reverse transcription quantitative PCR (RT-qPCR) was used to verify the hub genes in 14 pairs of TNBC paired tissues.

**Results**. In this study, we screened out 161 DEGs between 222 non-TNBC and 126 TNBC samples, of which 105 genes were up-regulated and 56 were down-regulated. These DEGs were enriched for 27 GO terms and two pathways. GO analysis enriched mainly in "cell division", "chromosome, centromeric region" and "microtubule motor activity". KEGG pathway analysis enriched mostly in "Cell cycle" and "Oocyte meiosis". PPI network was constructed and then 10 top hub genes were screened. According to the analysis results of the Kaplan-Meier survival curve, the expression levels of only NUF2, FAM83D and CENPH were associated with the recurrence-free survival in TNBC samples ($P < 0.05$). RT-qPCR confirmed that the expression levels of NUF2 and FAM83D in TNBC tissues were indeed up-regulated significantly.

**Conclusions**. The comprehensive analysis showed that NUF2 and FAM83D could be used as potential biomarkers for diagnosis and prognosis of TNBC.

Corresponding author
Dandan Zhou,
zhoudandancq@163.com

## INTRODUCTION

There were approximately 18.1 million new cancer cases worldwide in 2018, including 2.1 million cases of breast cancer (*Bray et al., 2018*). Breast cancer is the highest incidence among new morbidity and mortality in females with cancer (*Cao et al., 2019*). According
to variations in the expressions of the estrogen receptor (ER), progesterone receptor (PR) and human epidermal growth factor receptor 2 (HER2), breast cancer were defined as four major intrinsic molecular subtypes: luminal A, luminal B, HER2-positive and triple-negative breast cancer (TNBC) (*Sorlie et al., 2001*). TNBC is characterized by a lack of expression of the ER and PR as well as HER2 (*Serra et al., 2014*). TNBC that occurs mostly in premenopausal young women represents approximately 15–20% of all invasive breast cancers(*Foulkes, Smith & Reis-Filho, 2010*). TNBC is a highly heterogeneous disease, not only at the molecular level, but also in terms of its pathology and clinical manifestation. Its prognosis is worse than other types of breast cancer as well as the risk of death is higher (*Metzger-Filho et al., 2012*). Chemotherapy is currently the primary adjuvant treatment, due to the lack of effective molecular targets, it is not only insensitive to endocrine therapy and HER-2 targeted therapy, but also easily causes chemo-resistant (*Wein & Loi, 2017*). TNBC has become an intractable problem for clinical treatment.

Current researchers are focusing on personalized treatment based on the multi-gene assays (*Pan et al., 2019*). With the continuous development of high-throughput sequencing technology, bioinformatics analysis plays a key role in the diagnosis, prognosis and screening of tumors (*Goldfeder et al., 2017*; *Ma, Zhou & Zheng, 2020*). Many genes have been identified as signatures for diagnosis and prognosis of triple negative breast cancer (*Dai et al., 2019*; *Stovgaard et al., 2020*). A recent study found that CHD4-β1 integrin axis may be a prognostic marker for TNBC using next-generation sequencing and bioinformatics analysis (*Ou-Yang et al., 2019*). The computational analysis of complex biological networks could help research scholars identify potential genes related to TNBC (*Li et al., 2020*).

In this study, we first identified a group of differentially expressed genes (DEGs) associated with TNBC from the Gene Expression Synthesis (GEO) database. Then, based on bioinformatics analysis, three candidate genes related to TNBC diagnosis and prognosis were successfully identified. Finally, reverse transcription quantitative PCR (RT-qPCR) was used to verify the candidate biomarkers in TNBC tissues and adjacent tissues. The current research aimed to explore potential biomarkers that may be highly correlated with the prognostic and diagnostic value of triple negative breast cancer.

## MATERIAL AND METHODS

### Data source

Triple-negative breast cancer gene expression data sets in this study were obtained from the publicly available GEO databases (https://www.ncbi.nlm.nih.gov/geo/) (*Barrett et al., 2013*). Three independent data sets from GSE45827 (*Gruosso et al., 2016*), GSE38959 (*Komatsu et al., 2013*), GSE65194 (*Maire et al., 2013*) were included. GSE45827 consists of 100 non-triple-negative breast cancer (non-TNBC) samples and 41 TNBC samples, GSE65194 consists of 109 non-TNBC and 55 TNBC samples, both GSE65194 and GSE45827 are based on the platform GPL570 [HG-U133_Plus_2] Affymetrix Human Genome U133 Plus 2.0 Array. GSE38959 consists of 13 non-TNBC and 30 TNBC samples, and the platform is GPL4133 Agilent-014850 Whole Human Genome Microarray 4x44K G4112F. All of the data sets were available online.

A total of 14 TNBC patients were collected in Chongqing Traditional Chinese Medicine Hospital. All patients were diagnosed with triple negative breast cancer (ER-negative, PR-negative, HER-2-negative) by histopathological examination, excluding other malignant tumors and no important organ diseases, such as severe cardiovascular, liver disease as well as renal insufficiency. A total of 28 frozen tissue specimens contained 14 tumor tissues and 14 matched adjacent non-tumor tissues were obtained. All tissues were collected immediately after surgical resection, and snap-frozen in liquid nitrogen until RNA extraction. Clinical information was obtained for all patients by the investigator from medical records. The more detailed clinical information is shown in File S1. This study has been approved by the Chongqing Hospital of Traditional Chinese Medicine ethics committee and written informed consent was obtained from all patients.

## Data processing of DEGs

R software (v3.6.2; http://www.r-project.org) was used for bioinformatics analysis. First, the gene expression profiles of three data sets were downloaded by using GEOquery package. Subsequently, background adjustments were performed by using the dplyr package. Finally, we utilized log2 transformation to normalize the data using the limma package. The RobustRankAggreg package was used to screen the differentially expressed genes, using adjust $P$ value <0.01 and $|logFC| \geq 2$ as cut-off criteria. The VennDiagram package was used to present significant co-expression genes.

## GO enrichment and KEGG pathway analysis of DEGs

Gene ontology (GO) (*The Gene Ontology, 2019*) is a tool for annotating genes from various ontologies, including biological processes (BP), cellular components (CC), molecular functions (MF). The Kyoto Encyclopedia of Genes and Genomes (KEGG) (*Kanehisa et al., 2019*) is famous for "understanding the advanced functions and utility resource library of biological systems", KEGG pathway mainly presents intermolecular interactions and intermolecular networks. GO enrichment and KEGG pathway analysis for DEGs were performed through the DAVID database (v6.8; http://david.abcc.ncifcrf.gov/) (*Jiao et al., 2012*) with "after FDR" (corrected $P$-Value < 0.01, gene count $\geq$ 5) set as statistically significant. The ggplot2 package in R was used to visualize the GO functional enrichment results.

## Protein-protein Interaction (PPI) networks and hub gene analysis

The online STRING database (v11.0; https://string-db.org/) collects and integrates information on the correlation between known and predicted proteins from multiple species (*Szklarczyk et al., 2019*). PPI network analysis could systematically study the molecular mechanisms of disease and discover new drug targets. The DEGs screened previously were mapped via the STRING database. Subsequently, visual analysis of the PPI network was matched to Cytoscape (v3.7.2; https://cytoscape.org), and hub genes were analyzed with the Cytoscape plugin CytoHubba (*Chin et al., 2014*). The DMNC algorithm was used to identify the top 10 hub genes.

**Table 1  Primers sequence of target gene and internal reference gene.**

| Gene | Primers |
| --- | --- |
| NUF2 | Forward Primer: 5′-TACCATTCACCAATTTAGTTACT-3′ |
| | Reverse Primer: 5′-TAGAATATCAGCAGTCTCAAAG-3′ |
| FAM83D | Forward Primer: 5′-AGTTCCGAATCCTGTATGCC-3′ |
| | Reverse Primer: 5′-GCTCCTTGGACTGTGGTTT-3′ |
| CENPH | Forward Primer: 5′-CCTTATTTTGGGGAGTAAAGTCAAT-3′ |
| | Reverse Primer: 5′-ACAAATGCACAGAAGTATTCCAAAT3′ |
| GAPDH | Forward Primer: 5′-AGGTCGGTGTGAACGGATTTG-3′ |
| | Reverse Primer: 5′-GGGGTCGTTGATGGCAACA-3′ |

## Survival analysis

The Kaplan Meier plotter, an online survival analysis tool, could rapidly assess the effect of 54k genes on survival in 21 cancer types (http://kmplot.com/analysis/), including the effect of 22,277 genes on breast cancer prognosis (*Gyorffy et al., 2012*; *Gyorffy et al., 2010*). In this study, TNBC patients were only screened out based on the intrinsic sub-type (basis: $n = 879$). Probes of genes were selected "only JetSet best probe set" (*Li et al., 2011*). Recurrence-free survival (RFS) was selected for survival analysis of candidate hub genes, $P < 0.05$ was considered to be statistically significant.

## Validation of hub genes

RT-qPCR was used to further verify the mRNA expression of the candidate hub genes in TNBC tissues and adjacent tissues. Total RNA from TNBC patients' tissues was isolated by TRIzol reagent (Invitrogen, Carlsbad, CA, USA). Total RNA quantity was evaluated by a NanoDrop ND-1000 spectrophotometer (Thermo Fisher Scientific, Waltham, MA, USA). RNA was reverse transcribed into cDNA according to the instructions of the Takara kit (Takara Bio Inc., Japan). RT-qPCR reactions were performed using the SYBR Green PCR Master Mix System (Tiangen Biotech, Beijing, China). GAPDH was used as a control to compare the relative expression of NUF2, FAM83D and CENPH mRNA in 14 pairs of triple negative breast cancer paired tissues. Three replicate holes were performed for target genes in the RT-qPCR experiment, and the primer sequences are shown in Table 1. The primers of the target genes and the internal reference gene were synthesized by Sangon Biotech (Shanghai) Co., Ltd.

## Statistical analysis

Statistical analyses of this study were analyzed with R software v3.6.2 and GraphPad Prism 5.0. Two-tailed Student's $t$-test was used to test significance of differences between two groups, and $P < 0.05$ was considered statistically significant. The RT-qPCR results were calculated and evaluated using the $2^{-\triangle\triangle Ct}$ method.

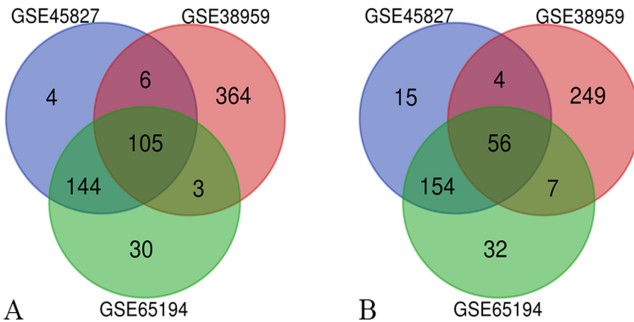

**Figure 1  Venn diagrams of the differentially expressed genes (DEGs).** Venn diagrams of the differentially expressed genes (DEGs) from three independent data sets (GSE45827, GSE38959, GSE65194). (A) a total of 105 up-regulated DEGs were identified and (B) 56 down-regulated DEGs were identified, using adjust *P* value < 0.01 and |logFC| ≥ 2 as cut-off criteria.

## RESULTS

### DEGs in non-TNBC and TNBC samples

Three series of matrix files, for a total of 222 non-TNBC samples and 126 TNBC samples, were selected to identify DEGs ($P < 0.01$, |logFC| ≥ 2). A total of 488 genes were identified after analyzing GSE45827, of which 259 genes were up-regulated and 229 genes were down-regulated. In gene chip GSE38959, 794 DEGs were identified, 478 genes were up-regulated, and 316 genes were down-regulated. And from GSE65194, 531 DEGs including 282 up-regulated genes and 249 down-regulated genes were identified. The Venn diagrams showed that a total of 161 DEGs overlapped, in which 105 genes were up-regulated and 56 genes were down-regulated (Fig. 1). The more detailed results are shown in File S2.

### GO and KEGG pathway analysis of DEGs

Next, we attempted to identify the biological function of the 161 common DEGs. GO enrichment and KEGG pathway analysis were performed through the DAVID database. Terms with matching the filter criteria were collected and grouped into clusters according to their membership similarities. As shown in Fig. 2, the top 5 functions for biological processes were as follows: cell division, mitotic nuclear division, chromosome segregation, sister chromatid cohesion and cell proliferation. The top 5 functions for cellular components were as follows: chromosome centromeric region, midbody, nucleus, condensed chromosome kinetochore and kinetochore. The molecular functions enriched were associated with microtubule motor activity, microtubule binding, ATP binding and protein binding. The KEGG analysis showed that the main enriched signaling pathways were related to the cell cycle and oocyte meiosis. The more detailed results are shown in File S3.

### PPI network construction and hub genes detection

In order to better understand which of these DEGs were most likely to be the central regulatory genes for TNBC, PPI network was constructed through the online STRING platform and Cytoscape software (Fig. 3A). Subsequently, according to the DMNC algorithm, the top 10 hub genes were screened through the cytoHubba and are sequentially
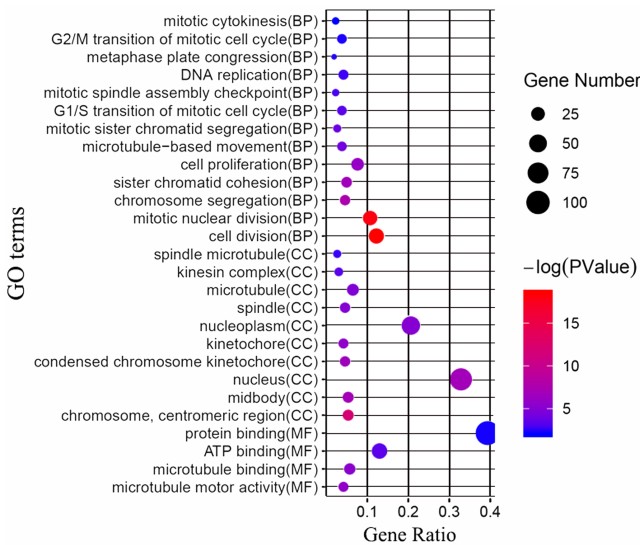

**Figure 2  GO enrichment analysis of the differentially expressed genes (DEGs).** GO enrichment analysis of the differentially expressed genes (DEGs). BP, biological processes; CC, cellular components; MF, molecular functions.

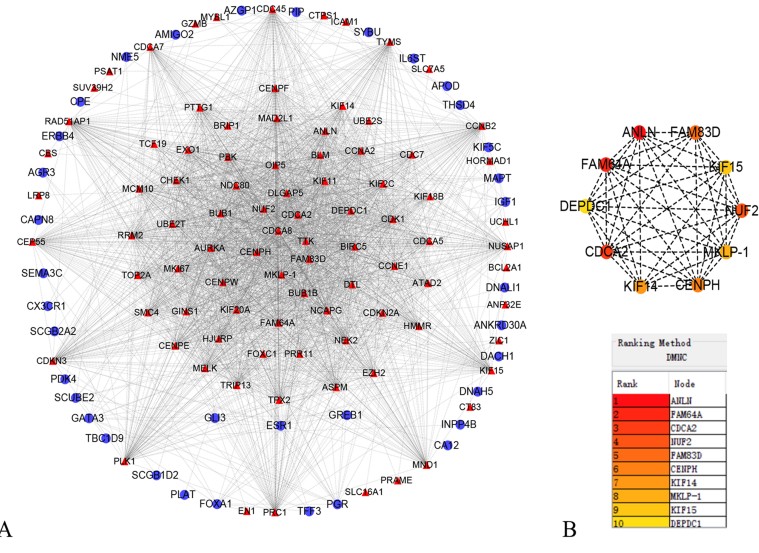

A                                                                                B

**Figure 3  Protein–protein interaction network of the differentially expressed genes (DEGs).** (A) Protein–protein interaction network of the differentially expressed genes (DEGs). Red color represents up-regulated genes, blue color represents down-regulated genes. (B) Identification of the top 10 hub DEGs by cytoHubba plugin. The rank is represented by different degrees of color (from red to yellow).

ranked as follows: ANLN, FAM64A, CDCA2, NUF2, FAM83D, CENPH, KIF14, MKLP-1, KIF15, DEPDC1 (Fig. 3B). The expression of 10 hub genes were all significantly increased in the PPI network. We initially speculate that 10 candidate hub genes may be related to tumor occurrence.

## Survival analysis and validation of hub genes

In order to examine whether the candidate hub genes expression levels were associated with the outcome of TNBC patients. Next, the correlation between these genes and the recurrence-free survival of TNBC patients were analyzed by the Kaplan Meier plotter. According to the analysis results of the Kaplan–Meier survival curve, we found that TNBC patients with higher expression levels of NUF2, FAM83D, CENPH have significantly decreased recurrence-free survival ($P < 0.05$), but not ANLN, FAM64A, CDCA2, KIF14, MKLP-1, KIF15, DEPDC1 ($P > 0.05$). More specific information about these survival-related hub genes is shown in Fig. 4.

Finally, we validated the expression levels of NUF2, FAM83D and CENPH in 14 pairs of triple negative breast cancer paired tissues by using RT-qPCR. Figure 5 showed that the expression levels of NUF2 and FAM83D were significantly higher in TNBC tissues than adjacent tissues ($P < 0.001$), but not CENPH ($P = 0.68$). Combined with the above analysis, we preliminarily concluded that NUF2 and FAM83D may be potential biomarkers to TNBC diagnosis and prognosis. The more detailed results are shown in File S4.

## DISCUSSION

TNBC is considered as an aggressive subtype of breast cancer. Compared with other types of breast cancer, TNBC is characterized by high malignancy rate, easier recurrence (*Dent et al., 2007*), and low survival rate (*Carey et al., 2006*). Despite advances in the targeted therapies of TNBC, including the approval of poly-ADP-ribose polymerase (PARP) and immune check-point inhibitors for the treatment of BRCA germ cell mutated breast cancers, there is still a lack of clinical evidence to evaluate their efficacy for TNBC patients (*Vagia, Mahalingam & Cristofanilli, 2020*). Therefore, it is necessary to identify effective molecular therapeutic targets for TNBC.

In the present study, we screened out 161 DEGs between 222 non-TNBC and 126 TNBC samples by analyzing three datasets, of which 105 were up-regulated and 56 were down-regulated. The GO enrichment analysis and KEGG pathways showed that the screened DEGs were enriched for 27 GO terms and 2 pathways. To further investigate the interrelationship of 161 DEGs, PPI network was first constructed and then 10 top hub genes were screened out, including ANLN, FAM64A, CDCA2, NUF2, FAM83D, CENPH, KIF14, MKLP-1, KIF15, DEPDC1. The analysis results of the Kaplan–Meier survival curve showed that the expression levels of NUF2, FAM83D and CENPH were associated with the recurrence-free survival in TNBC samples ($P < 0.05$). Finally, we found that the expression levels of only NUF2 and FAM83D did increase significantly in TNBC tissues by using RT-qPCR.

NUF2 is an essential component of the kinetochore-associated NDC80 complex, which plays a regulatory role in chromosome segregation and spindle checkpoint activity (*Liu et al., 2007*; *Zhang et al., 2015*). Several studies have shown that NUF2 was associated with the development of multiple cancers. The results showed that the expression of NUF2 was associated with poor prognosis in patients with colorectal cancer (*Kobayashi et al., 2014*) and oral cancer (*Thang et al., 2016*), which may be related to the regulation of tumor cell

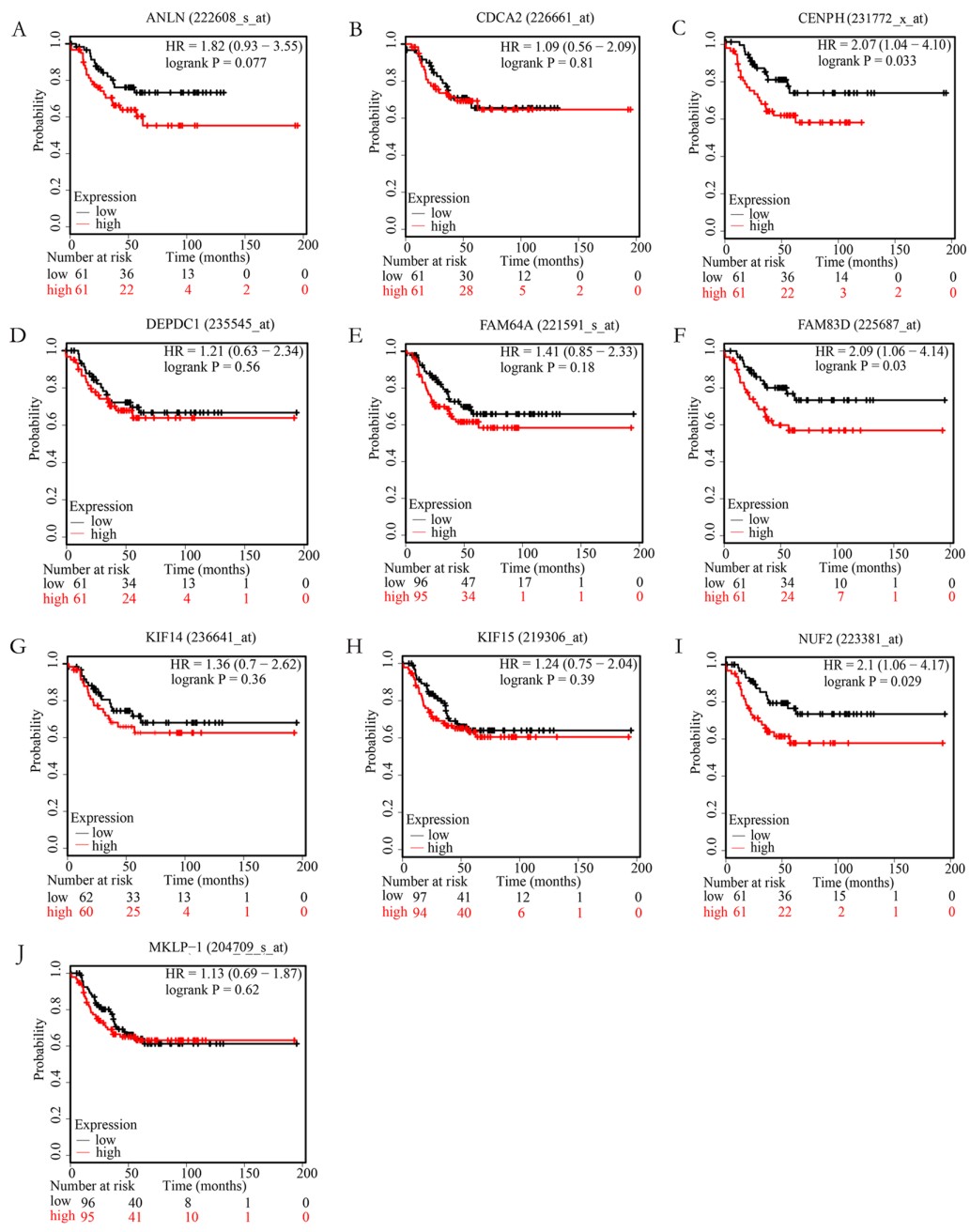

**Figure 4** **The correlation of 10 hub genes expression levels with the recurrence-free survival of triple-negative breast cancer (TNBC) samples.** TNBC patients with higher expression levels of CENPH (C), FAM83D (F), NUF2 (I) have significantly decreased recurrence-free survival ($P < 0.05$), but not ANLN (A), CDCA2 (B), DEPDC1(D), FAM64A (E), KIF14 (G), KIF15 (H), MKLP-1 (J) ($P > 0.05$).

apoptosis involved in the NUF2. Sugimasa H et al (*Sugimasa et al., 2015*) demonstrated that the NUF2 gene could be directly trans-activated by the heterogeneous ribonucleoprotein K (hnRNP K), and that the hnRNP K-NUF2 axis affected the growth of colon cancer cells by participating in processes of mitosis and proliferation. Recent studies have shown

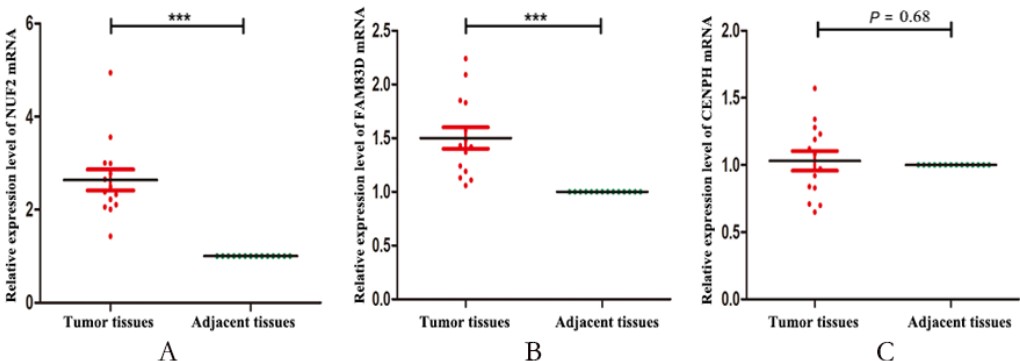

**Figure 5 The relative expression levels of NUF2, FAM83D and CENPH mRNA in 14 pairs of triplenegative breast cancer (TNBC) paired tissues.** The mRNA expression levels of NUF2 (A) and FAM83D (B) increased significantly in most TNBC lesions compared with para-adjacent tissues, but not CENPH (C). *** $P < 0.001$.

that NUF2 was also closely related to breast cancer. Xu et al (*Xu et al., 2019*) confirmed that NUF2 was indeed up-regulated in breast cancer tissue by bioinformatics analysis and RT-qPCR assay, and that NUF2 may regulate the carcinogenesis and progression of breast cancer via cell cycle-related pathways. However, the expression level changes of NUF2 in triple-negative breast cancer have not yet been studied. In this study, we found that the expression level of NUF2 was higher in triple-negative breast cancer than in non-triple negative breast cancer and TNBC patients with higher NUF2 expression level had significantly reduced the recurrence-free survival. GO enrichment analysis shows that NUF2 is mainly involved in cell division, mitotic nuclear division, chromosome segregation and sister chromatid cohesion, their dysregulation impact significantly on development of cancer (*Bakhoum et al., 2018*; *Guo et al., 2013*; *Lopez-Lazaro, 2018*). Based on the above analysis, we speculate that NUF2 plays an important role in tumor progression, and NUF2 may be serve as a biomarker for diagnosis and prognosis of triple-negative breast cancer. Certainly, the specific molecular mechanism of NUF2 expression level changes in TNBC still need to be further studied.

FAM83D belongs to the FAM83 family, which could regulate cell proliferation, growth, migration and epithelial to mesenchymal transition (*Li et al., 2018*; *Santamaria et al., 2008*). The studies have found that FAM83D could not only affect cell proliferation and motility through the tumor suppressor gene FBXW7 (*Mu et al., 2017*) or ERK1/ERK2 signaling cascade (*Wang et al., 2015*), but also affect breast cancer cell growth and promote epithelial cell transformation through MAPK signaling (*Cipriano et al., 2013*; *Cipriano et al., 2014*; *Lee et al., 2012*). The expression of FAM83D was significantly increased in primary breast cancer and the high expression level of FAM83D was closely related to the adverse clinical outcomes and distant metastasis in breast cancer patients (*Wang et al., 2013*). In our study, we found that the expression of FAM83D was significantly increased in TNBC patients and TNBC patients with higher FAM83D expression level had significantly reduced the recurrence-free survival. GO enrichment analysis shows that FAM83D is mainly involved

in cell division, mitotic nuclear division and cell proliferation, their dysregulation have a major impact on the development of cancer (*Bakhoum et al., 2018*; *Lopez-Lazaro, 2018*; *Wu et al., 2019*). We speculated that FAM83D might play a role in the progression and prognosis of triple-negative breast cancer.

Centromere protein H (CENP-H) is a component of the kinetochore and plays an essential role in mitotic processes (*Lu et al., 2017*), accurate chromosome segregation (*Zhu et al., 2015*) as well as appropriate kinetochore assembly (*Zhao et al., 2012*). Many studies have shown that CENPH is closely associated with human cancers, including colorectal cancer (*Wu et al., 2017*), renal cell carcinoma (*Wu et al., 2015*), non-small cell lung cancer (*Liao et al., 2009*) as well as breast cancer (*Walian, Hang & Mao, 2016*). However, there is no current evidence on the correlation between CENPH and triple negative breast cancer. In this study, we found that there is no significant correlation between the mRNA expression of CENPH and triple negative breast cancer.

It is worth noting that protein-coding genes are not the sole drivers for cancer. Breast cancer is also related to the expressions of non-coding RNAs, include repetitive DNA (*Yandim & Karakulah, 2019*), transposable element (*Karakulah et al., 2019*), micro RNA (*Aslan et al., 2020*) and Long non-coding RNA (*Riahi et al., 2020*),etc. In this study, we have found that the expressions of NUF2 and FAM83D are associated with triple-negative breast cancer. Next, we will further investigate whether the expression changes of NUF2/FAM83D in triple-negative breast cancer are caused by non-coding RNA.

## CONCLUSION

In summary, we firstly demonstrated that the mRNA levels of NUF2/ FAM83D have changed significantly in TNBC tissues compared to adjacent tissues. The mRNA expression levels of NUF2/FAM83D are significantly up-regulated in TNBC tissues. NUF2/FAM83D might serve as potential molecular biomarkers for diagnosis and prognostic indicators of TNBC. However, the functional mechanisms of NUF2 and FAM83D in TNBC patients are still to be further studied, including the expression of their protein levels and their relationship with the clinical characteristics of TNBC patients and so on. Therefore, we still need to do more experiments before clinical trials.

### Funding
The authors received no funding for this work.

### Competing Interests
The authors declare there are no competing interests.

### Author Contributions
- Xiuming Zhai and Dandan Zhou conceived and designed the experiments, performed the experiments, analyzed the data, prepared figures and/or tables, authored or reviewed drafts of the paper, and approved the final draft.

- Zhaowei Yang performed the experiments, prepared figures and/or tables, authored or reviewed drafts of the paper, and approved the final draft.
- Xiji Liu and Zihe Dong analyzed the data, prepared figures and/or tables, authored or reviewed drafts of the paper, and approved the final draft.

## Human Ethics

The following information was supplied relating to ethical approvals (i.e., approving body and any reference numbers):

Chongqing Traditional Chinese Medicine Hospital granted Ethical approval to carry out the study within its facilities.

## Data Availability

The raw measurements are available in the Supplemental Files.

## Supplemental Information

Supplemental information for this article can be found online at http://dx.doi.org/10.7717/peerj.9975#supplemental-information.

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
