# Peer review of "Identification of NUF2 and FAM83D as potential biomarkers in triple-negative breast cancer"

_PeerJ, doi:10.7717/peerj.9975_

## Round 0.1 · original submission · Major Revisions

- For further consideration of the manuscript, I would like to invite the authors to implement all referee comments.
- As suggested by reviewers, the manuscript needs to be edited by a native-equivalent speaker to increase the readability.
- Breast cancer also shows changes in the expressions of noncoding RNAs, (e.g. PMID: 31536958 and PMID: 31824778) and one should also mention that protein-coding genes should not be the sole drivers. The authors include these in the revised version of the discussion.
- Figure 3 is not clear to me. Gene symbols cannot be read properly and the scroll bar should be removed from 3B. This figure should be improved in the revised manuscript.
- In Figure 2, "Go_term" should be replaced with "GO terms". It is also not clear to me what is "EnrichmentScore" and how it was calculated?

Reviewer 1 ·

Basic reporting

In the present article, entitled “Identification of NUF2 and FAM83D as novel biomarkers in triple-negative breast cancer”, authors have performed the computational and experimental analysis to identify the effective biomarkers to triple-negative breast cancer (TNBC) diagnosis and prognosis. It is good that after analyzing the GEO data sets, authors have experimented to validate the data analysis results.

However, some important points need to be clarified or fixed

1. There are many vague sentences which should be edited, such as
a. “GO enrichment analysis contain 13 biological processes, 10 cellular components and 4 molecular functions”
b. “...deal with various confusing biological issues”.

2. Is there any novelty in the present work, which was not reported earlier?

Experimental design

1. DEGs and survival analysis were done in different data sets. Will this create any technical error in the results? Authors should justify
2. Why was Recurrence-free survival (RFS) presented? Why not overall survival?

Validity of the findings

1. How are the GO and KEGG pathway analysis results linked to TNBC development?
2. PPI network was constructed from publicly available data, the network should be validated and should be compared with the random network model [Molecular Genetics and Genomics volume 294, pages931–940(2019)].
3. The authors should also calculate the expression correlation between hub genes. If there are higher correlation, then those hub genes are probably involved in similar biological function.

Reviewer 2 ·

Basic reporting

no comment

Experimental design

no comment

Validity of the findings

no comment

Additional comments

Zhai et al. submitted the manuscript” Identification of NUF2 and FAM83D as novel biomarkers in triple-negative breast cancer” they use multiple bioinformatics methods to identify two genes (NUF2 and FAM83D) as potential biomarkers for TNBC. Their correlation was validated by real-time PCR. Overall, the manuscript is generally well-written and clearly presented, figures are relevant, high quality. However, there are some typos and phrases that need rewriting. In addition, it will be more convincing if the authors present more experimental evidence to support the correlation between NUF2/FAM83D and TNBC. Here I have some minor comments and suggestions that may be helpful to improve the manuscript.
1. There are some sentences that were grammatically incorrect.
Line 62 “Many genes have been found could be used as signatures for TN breast cancer diagnosis and prognosis”; Line150-151“The Venn diagrams showed that a total of 161 genes were co-expression”; Line 228-229 “Certainly, the specific molecular mechanisms by which NUF2 mediates TNBC carcinogenesis still need to be further study.”.
2. Some other phrases, albeit grammatically correct, were scientifically confusing.
Line 67 “deal with various confusing biological issues”; Line 72-73 ” This comprehensive analysis may provide a meaningful contribution to the targeted treatment of triple negative breast cancer”; Line 151” clearly up-regulated”; Line 221-222 “However, the role of NUF2 in triple negative breast cancer has not been conducted”
3. Line 155”161common” space should be place between “161” and “common”. So does line 147 ” 259genes”
4. Please define “BP” in line 158, and “CC” in line 160 and “MF” in line 161.
5. There are two types of microarray in the three datasets, GSE65194 and GSE45827 are Affymetrix arrays while GSE38959 are Agilent array (line 76-84), please indicate more details about data normalization (Line 96).
6. In section “DEGs in non-TNBC and TNBC samples”, Please submit the full list of differential expression genes (at least 161 common DEGs) as supplemental files.
7. The authors validated the correlation between NUF2/FAM83D and TNBC only by real-time PCR, however, it still needs more evidence to make a conclusion that NUF2 and FAM83D are TNBC biomarkers. It will be very helpful if you can present histological evidence (you already have clinical samples, line 85) OR in vitro assay, for example, you may confirm the function of NUF2 and FAM83D on TNBC cell lines (if possible).

Reviewer 3 ·

Basic reporting

Are NUF2 and FAM83D really a novel biomarker? Novel means that these genes have not identified as a biomarker yet but in previous studies eg. https://pubmed.ncbi.nlm.nih.gov/31140425/, https://www.spandidos-publications.com/ijmm/44/2/390, https://www.ncbi.nlm.nih.gov/pmc/articles/PMC4823109/
, it has been shown that the prognostic significance of these genes. Suggest removing the word ‘novel’.

In the abstract, the author mentioned the need of identifying ‘effective biomarkers’ to diagnose or determine the prognosis of TNBC. However, this study merely reports the potential biomarkers. Further studies are needed to determine if the reported biomarkers are useful for diagnosis/prognosis. Consider revising some of the words/sentences in the manuscript as some might sound like an overstatement.

Experimental design

In line 88: the author mentioned ‘no important organ disease’. Please specify the example of organ diseases as the exclusion criteria.

Line 105-6: is the sentence a repetition? Shouldn’t it be ‘intermolecular interactions and intermolecular networks’?

Line 135-6: Why the need for the experiment to be repeated (independently) for more than 3 times? 14 pairs of TNBC samples were used, do you mean experimental replicates were done more than 3 times. If yes, why is this so because the replication level of your experiments will have an impact on the statistical tests.

Validity of the findings

Although the results from this study show no correlation between CENPH and TNBC, the author may discuss or elaborate more on the role of CENPH in other types of breast/other cancers.

The summary in line 244 is quite weak and inconclusive.

The language used in this manuscript needs to be improved; there are a few grammatical errors throughout the manuscript.

The manuscript lacks new information. My suggestion would be to further study the functional role of each potential gene identified in this study to reach a fair conclusion.

---

## Round 0.2 · accepted · Accept

The authors successfully implemented the criticisms and comments raised by the reviewers during the revision period. I congratulate the authors.

Reviewer 1 ·

Basic reporting

Authors have answered all comments

Experimental design

Authors have answered all comments

Validity of the findings

Authors have answered all comments

Additional comments

Authors have answered all comments

Reviewer 2 ·

Basic reporting

no comment

Experimental design

no comment

Validity of the findings

no comment

Additional comments

Zhai et al. submitted the revised manuscript ”Identification of NUF2 and FAM83D as Potential Biomarkers in Triple-negative Breast Cancer”, the revised manuscript has great improvement, in which the authors corrected the typos and rewrite some phrases, and provided more detailed data to support their conclusion. They addressed all my concerns. I would like to recommend the manuscript to “accept”.